# Pineapple Agro-Industrial Biomass to Produce Biomedical Applications in a Circular Economy Context in Costa Rica

**DOI:** 10.3390/polym14224864

**Published:** 2022-11-11

**Authors:** Valeria Amores-Monge, Silvia Goyanes, Laura Ribba, Mary Lopretti, Manuel Sandoval-Barrantes, Melissa Camacho, Yendry Corrales-Ureña, José Roberto Vega-Baudrit

**Affiliations:** 1Sede de Occidente, Universidad de Costa Rica, San José 2060, Costa Rica; 2Departamento de Física, Facultad de Ciencias Exactas y Naturales, Universidad de Buenos Aires, Buenos Aires 1053, Argentina; 3Instituto de Física de Buenos Aires (IFIBA)CONICET, Universidad de Buenos Aires, Buenos Aires 1428, Argentina; 4Dirección de Materiales Avanzados, Áreas del Conocimiento, INTI-CONICET, Buenos Aires 5445, Argentina; 5Departamento de Técnicas Nucleares Aplicadas en Bioquímica y Biotecnología, CIN, Facultad de Ciencias, Universidad de la República, Montevideo 11400, Uruguay; 6School of Chemistry, Universidad Nacional, Campus Omar Dengo, Heredia 3000, Costa Rica; 7Laboratorio Nacional de Nanotecnología LANOTEC-CeNAT, San José 1200, Costa Rica

**Keywords:** pineapple, wastes, biorefinery, COVID-19, circular economy, biomedical

## Abstract

Pineapple is a highly demanded fruit in international markets due to its unique appearance and flavor, high fiber content, vitamins, folic acid, and minerals. It makes pineapple production and processing a significant source of income for producing countries, such as Costa Rica. This review collects bibliographic information dating back to the beginnings of pineapple production in Costa Rica to the state of the market today. It details the impacts of its production chain and proposes a biorefinery as a solution to environmental problems. Besides the potentiality of new sustainable markets to contribute to the post-COVID-19 economy in Costa Rica is highlighted. The general characteristics of pineapple by-products -cellulose, hemicellulose, lignin, and other high-value products like bromelain y saponin- are described, as well as the primary processes for their ex-traction via biorefinery and main applications in the medical field. Finally, a brief description of the main works in the literature involving modeling and simulation studies of pineapple by-products properties is included.

## 1. Introduction

Pineapple production in Costa Rica has been one of the largest sources of income and work since the 1980s. In the 1990s, the export of fresh pineapple began, mainly to the United States and Europe, and increased to a considerable extent once the country’s pineapple sector received several international certifications to affirm such aspects as product quality, environmental protection, innocuity, and social responsibility, among others [1].

Large-scale pineapple production began by the Pindeco SA company, which was in charge of creating a product that covers as many cultivation hectares as possible through dependence on technological machinery and the use of chemical products, so much so that they exceeded the number of chemicals used in banana cultivation [2].

There has been such a substantial expansion in production that, according to a comparative study, Costa Rica increased from 24,000 tons in 1994 to producing nearly 3 million tons in 2014, which in turn, directly and indirectly, generated 32,000 jobs by 2017 [3]. By 2020, the total of pineapple exported worldwide from Costa Rica was 1,993,767.56 tons, where the leading destinations for this product were the United States, the European Union, the United Kingdom, Russia, and Turkey (see Figure 1) [4].

Due to the high exportation and demand for pineapple, several modifications had to be made. For example, pineapple is grown in three different areas to achieve the highest possible amount of produced pineapple, which corresponds to the North Sea, Atlantic Ocean, and the Pacific Ocean; notably, a specific variety of exported fruit, referred to as “Golden Pineapple”, has generated production that improves the quality using pesticides and fertilizers [5].

The use of this type of chemical product, soil overexploitation, and inappropriate use of waste have had a negative impact on the environment, society, and human health. An example of this is chemical contamination of the aquifers near the crops that are a source of water for the nearby villages, which threatens the residents’ health; moreover, the inappropriate use of waste led to an overpopulation of the stable fly (i.e., *Stomoxys calcitrans*), which is associated with problems in cows such as malnutrition and death, among others, and a decrease in species of monkeys in the northern part of the country due to extending the use of land for pineapple cultivation [2].

While the exact amount of waste produced per one-hectare crop is unknown, an estimate indicates that 300 metric tons are generated per hectare, which is alarming considering that there are currently more than 43,000 hectares dedicated to pineapple cultivation. For this reason, the need for biorefinery is evident; this practice consists of the transformation of biomass through biochemical or thermochemical processes to achieve a second use for waste matter and ensure all resources are utilized via renewable energy, thereby entering a circular economy system [6].

Biorefinery generates many value-added products that can be useful in several industrial areas, which can be obtained through different methods. One example of this practice is the use of solid-state fermentation to produce biofuels (i.e., bioethanol) to replace fossil fuels and lessen the use of forest biomass (i.e., reduced deforestation); these biofuels are primarily composed of materials with lignocellulose and are considered to be economically profitable and ecologically responsible [7]. There are also various medical products that can be created from the biomass of agricultural products, e.g., antibiotics from rice husks or peanuts. Studies have indicated that products such as oxytetracycline are obtained through solid-state fermentation in a *Streptomyces rimosus* culture; extracellular *rifamycin B*, an antibiotic used in the treatment of tuberculosis and leprosy, can be produced with the use of *Amycolatopsis mediterranei* [7].

The circular economy resulting from the creation of value-added products from pineapple waste biorefinery may help to mitigate the negative economic impact of the COVID-19 pandemic. This worldwide event caused severe problems in the economy of many countries, including that of Costa Rica, which mainly was affected at the national level by the paralysis of many productive activities that significantly affected production, exportation, jobs, and the income of the country and its citizens [8]. According to PROCOMER data, 2020 was the lowest income due to exportations in the last five years (2017–2021). Lignocellulosic materials can produce other materials with high added value, such as crystalline nanocellulose and dextrans, for clinical uses, as well as energy production, among others. Likewise, the pineapple circular economy can be promoted and even con-tribute to creating companies that contribute to the country’s economy [4].

For this reason, this study aims to compile bibliographic information dating back to the beginning of pineapple production from Costa Rican institutions such as the National Chamber of Pineapple Producers and Exporters—CANAPEP and the Ministry of Agriculture and Livestock—MAG; the benefits, environmental impact, and waste thereof; the most-used biorefinery treatments; the main components obtained from the biorefinery process, such as lignin, cellulose, and hemicellulose; the applications of these by-products in the pharmaceutical industry; and the contribution thereof to the circular economy in the post-COVID-19 economy for pineapple producing countries.

## 2. Discussion

### 2.1. Pineapple Production

Pineapple (i.e., *Ananas comosus*, Bromeliaceae family) is a fruit that is native to South America, specifically Brazil, Paraguay, and Argentina. It was discovered on Christopher Columbus’s second visit to this area, and consumption of pineapple spread during the 16th century due to trade along the Spanish and Portuguese sea lanes. Cultivation of this fruit can take between 14 and 20 months and involves three stages: planting and growth of shoots, flowering and harvesting, and the production of new shoots for future plantations [9].

Worldwide, pineapple cultivation has been very successful. There is a high demand for this fruit in the international market, mostly due to its flavor. Its high fiber content and potassium, iodine, carbohydrates, fiber, and vitamins A, B, and C, have prompted recommendations to consume pineapple to treat such diseases as fluid retention due to its diuretic effects, hypertension, cholesterol, anemia, poisoning due to its purifying action, immune system disorders due to its ability to strengthen defenses and assist in the formation of both red and white blood cells, cardiovascular problems, and obesity, among others [10,11].

Due to its tremendous success in the international market, pineapple cultivation has become a global industry. According to FAO, 2019 exports rose to 3.1 million tons, mainly due to the increase in Philippines exports and importations into China. Costa Rica is currently the largest producer and exporter of pineapple, followed by the Philippines and Brazil [12].

It should be noted that there is a variety of pineapples, each of which has different characteristics that make them widely traded. One variety is the Smooth Cayenne, which has a high yield, good potential for conservation, and organoleptic qualities; this pineapple is sensitive to problems such as black spots and other parasites, however, which can affect the overall quality. It is why MD 2, or the “Golden Pineapple,” was developed in the 1980s; this variety has excellent preservation properties, flavor, color, and minimal sensitivity to parasites and discoloration, and it is the most popular variety in commerce and exportation. There are several other widely distributed varieties, including but not limited to Queen Victoria, Sugar Loaf, and Champaka [9].

As was already mentioned, Costa Rica is the largest global producer and exporter of pineapple, followed by Philippines and Brazil. As mentioned before, pineapple exportation began in Costa Rica in the 1980s with the integration of the Pineapple Development Company (i.e., Pindeco), which led to increased production and higher crop yield [13]; this occurred with the use of imported technology and the creation of dependence on chemical inputs, such as herbicides, fungicides, insecticides, and fertilizers [14]. Production began to increase significantly: 24,204 tons of pineapple were produced in 1984, and this increased to 651,000 tons by 1998 [3]; production expanded to plantations in the Huetar Norte, the Atlantic Zone, and throughout the Central and South Pacific area. During this period, pineapple production began to gain additional relevance because of reducing poverty in rural sectors through the generation of direct and indirect jobs; there were 4200 direct jobs in 1995, and these grew to more than 31,000 in 2015 [1].

The Costa Rican pineapple production process (Figure 2) begins with cultivation, which requires the use of agrochemicals, machinery, fuels, and water; this process results in pineapple that can be exported and product that remains for local consumption. In the case of exportation, there is a packaging process before this fruit is exported to different countries and continents, and the United States and the European Union are the main importers of pineapple [5].

There are currently more than 65,000 hectares devoted to this crop in Costa Rica, which amounts to approximately 1.29% of the total national territory, which are located in four areas—the Huetar Norte Region, the Central Pacific Region, the Huetar Caribe Region, and the Brunca Region—the three main cantons of which are San Carlos, Los Chiles, and Buenos Aires [15]. Of the pineapple grown in the country, 90% is exported, and the remaining 10% is sold locally. The Monte Lirio, Cayena Lisa, Champaca, and MD 2 (i.e., “Golden Pineapple”) pineapple varieties are grown in Costa Rica; the MD 2 variety is considered to be the most important for export due to its high quality, which presents a high degree of brix and weighs as much as two kilograms [16].

Most Costa Rican pineapple exports are sent to the United States and the European market. Costa Rica also exports to the Central American, Middle Eastern, and Asian sec-tors [1]. Also, the pineapple was exported to China for the first time in 2017, which resulted in increased business on the Asian continent [17].

The main destinations detail of Costa Rican pineapple exportation in the first year of the pandemic outbreak are shown in Figure 1. The USA is the most important market, with 51% of the total exports. The European Union is the second most important market, with almost 36% of the total distributed in several countries. As combined consumers, the United Kingdom, Russia, and Turkey represent more than 10% of Costa Rican pineapple exportations. Finally, the rest of the countries represent 2.23% of the exports (Table 1).

### 2.2. Agricultural Waste

It is estimated that more than 4000 tons of solid waste are produced each day in Costa Rica. In general, this waste has been poorly handled; only 78% is collected by the municipalities and taken to sanitary landfills, and the location of the remainder of the waste and the environmental impact thereof is unknown [18]. In the specific case of agro-industrial waste, some is generated from industrial activity and processes, including the transformation of agricultural and livestock materials [19]. The number of residues from these processes have significantly increased due to the large-scale production that is demanded by the country’s agro-industrial sector, which also requires 81,000 hectares for coffee crops, 64,000 hectares for sugar cane, 53,000 hectares for bananas, and 40,000 hectares for rice, among others [20]. These requirements have resulted in several harmful impacts on the environment and society, forcing companies to be more sensitive to the effects of their activity and seek solutions for problems resulting from the mismanagement of solid waste [21].

The pineapple industry has not been the exception to these problems due to residues produced in the area of cultivation [22]. In anaerobic conditions, it takes approximately 13 months for the residues generated during pineapple cultivation to decompose. The length of time needed to continue cultivating and resulting in the presence of pests that cause diseases in pineapple crops are unfavorable factors for which various waste management solutions have been sought. The waste was initially burned, and while this was a low-cost option, it led to other environmental impacts. The same results occurred with alternative solutions, such as crushing and applying herbicides, the use of which are not recommended in the production of organic products; a proposed alternative for this was the use of decomposing microorganisms, which allow for improved soil mineralization and do not generate environmental problems [23].

Similarly, pineapple crops are associated with the generation of pesticide residues due to the widespread use of pesticides in this agricultural practice; this has resulted in multiple cases of pesticide contamination, mainly in bodies of water such as aquifers that provide drinking water to communities located near areas where pineapple is produced. As a result, several hydrogeological studies were conducted to demonstrate vulnerabilities; in the Guácimo and Pococí cantons, specific actions were taken to solve the problem with drinking water, resulting in the construction of two new aqueducts [24].

Furthermore, a large amount of waste is also generated in the pineapple processing industry because of pineapple by-products—such as pineapple juice, concentrated juice, dehydrated pineapple, and cut pineapple (also known as “canned pineapple”)—which represent a large part of the exports. These by-products generate a tremendous amount of waste because they only use parts of the cultivated pineapple and render the rest of the product unusable [25]. For this reason, agro-industrial waste is classified into two categories: agricultural waste and industrial waste, as is shown in Figure 3.

Agricultural waste includes waste generated when crops are harvested, such as leaves, stems, seeds, pods, and stems; additional crop processes that produce molasses, husk, bagasse, seeds, pulp, and stubble are also included in this category. Industrial waste includes material produced when creating the by-products of a fruit or vegetable, such as cellulose, hemicellulose, lignin, humidity, ashes, carbon, nitrogen, etc. that remains after juices and canned goods are created; these waste products are beneficial in the creation of other products, such as biofuels [7].

In general, the pineapple plant is composed of eight parts, the names of which correspond to the whole plant—the stubble, crown, shell, flesh, stem, roots, leaves, and heart—and all create a significant amount of lignocellulosic biomass (i.e., lignin, cellulose and hemicellulose); however, as is shown in Figure 4, most of this biomass is discarded after the pineapple, and the by-products mentioned above are produced [26].

### 2.3. Biorefinery and the Importance Thereof for Costa Rica and Other Pineapple Producing Countries

Biorefinery is a process with several definitions, which can be summarized in the use of all raw materials to create value-added products, by-products, and energy. General concept of biorefineries is typically composed of feedstock/raw materials; platforms; processes; and products. In this case, using pineapple wastes to create value-added products, by-products, and energy. Biorefinery is similar to the oil refinery (Figure 4) in that the objective is the use of all raw materials through a treatment similar to oil fractionation, wherein different products of each fraction are obtained [27]. Biorefinery procedures were devised to address the environmental impacts caused by the excessive use of fossil fuels, the mismanagement of agro-industrial waste, and industrial wood waste. This type of activity is intended to generate the least possible amount of non-renewable natural resources and reduce the use of fossil fuels to minimize possible effects on the environment [28].

The practice of reusing lignocellulosic biomass began in the 1930s when William Jay Hale envisioned the creation of commercial products from carbohydrates originating in the agro-industrial sector. The biomass produced from photosynthesis results from the reaction between carbon dioxide and water and is mainly composed of three biopolymers: cellulose, hemicellulose, and lignin. It should be noted that the composition and proportions of each biopolymer depend on the type of biomass and the origin thereof [29].

Multiple notable studies have been conducted in Costa Rica on biomass to create new value-added products. Many of these stand out because of different methodologies and various biomass components to determine different applications in various productive areas. Most research related to a biorefinery in Costa Rica has focused on the reuse of pineapple stubble due to the high demand for pineapples in the country and the environmental impact of production. Even though a pineapple biorefinery process to create biofuels has been proposed as a strategy to end oil dependence, not many companies have implemented it, so future studies are needed to explore the implementation of this process, and it is essential that aspects of the impact, carbon footprint, and life cycle of the variety of products are measured [6].

Despite not being implemented, the start and growth of this process was taken into account in the National Bioeconomy Strategy Costa Rica 2020–2030, in which objectives were proposed to transform Costa Rica into a model country in the development of sustainable resources that would make a bioeconomy one of the pillars of Costa Rica’s production transformation and promote convergence of the country’s wealth and biological resources. These objectives were intended to be fulfilled by structuring the country along five main axes: rural development, biodiversity, and development; residual biomass biorefinery; an advanced bioeconomy; an urban bioeconomy; and green cities [30].

Specifically, the main objective of the residual biomass biorefinery axis was to promote the development of new productive activities to ensure the full use and valorization of residual biomass from agricultural, agro-industrial, forestry, and fishing processes; with this objective, the Costa Rican government seeks to position the country as a leader in the field of biorefinery concerning the total use of residual biomass. This goal is expected to be achieved in conjunction with a large amount of research conducted by the National Nanotechnology Laboratory–Lanotec in Costa Rica. This area is supported by Executive Decree No. 36,567 by Science and Technology Ministry MICIT, which declared that nanotechnology research and the applications thereof are of public interest [30]. This decree was created to support and promote the development of research in the area of nanotechnology and nanomaterials (e.g., nanocellose), so public and private institutions are both motivated by science and technology innovations to achieve greater understanding and environmental sustainability, as well as to strongly promote economic and social development in Costa Rica [31].

### 2.4. Circular Economy

The circular economy, which consists of a sustainable economic model intended to maintain the efficiency of products in the productive market to the greatest possible degree and avoid generating waste, is closely related to the previous topic [32]. To achieve these goals, the circular economy is premised on three fundamental principles—the preservation and improved usage of natural capital, optimized yields of resources, and the promotion of efficient systems—all of which can be fulfilled by controlling finite reserves, rethinking production, and recycling systems that enable materials to continue circulating in the economy and reduce the damage to the environment caused by different sectors of the economy [33]. In general, the circular economy includes five fields of action: extraction, which is a process in which raw materials are obtained that attempts to avoid an environmental impact; transformation, wherein raw materials go through the industrial processes to create products in the most sustainable manner; distribution to customers, which seeks to ensure product traceability; the usage phase, wherein the customer has a product that ideally has the least possible impact on energy usage; and waste recovery for a product that ensures it can be recycled, and additional value-added products can be created (Figure 5) [34].

Economists have identified five main characteristics of the circular economy—waste-free production, increased resilience, the use of energy from renewable sources, thinking about the system, and a cascading effect—the last characteristic refers to the fact that several systems are not linear, and adding value to some products cascades to other applications [35]. This type of economy is a radical change from current systems because many fail to consider the impact of the massive amount of waste generated by companies day after day, which not only affects the environment but also has an impact on the climate and human health [36].

### 2.5. Main Raw Materials from Pineapple Biomass

As was mentioned earlier in this text, the lignocellulosic biomass obtained from pineapple agro-industrial residues is mainly composed of cellulose, hemicellulose, and lignin; a compilation of information for these three compounds is presented below, with a specific focus on treatments and pretreatments that can be employed to obtain and purify them, as well as the most common applications of each.

#### 2.5.1. Lignin

Lignin is one of the most abundant biopolymers and is exclusively found in vascular plants, where it fulfills its function by providing structural integrity and mechanical resistance [37,38]. From pineapple peel wastes, lignin is in the range of 15–30% by dry weight [39]. Lignin is formed by a reaction of photosynthesis in cell walls, and it has a resin role because it fills spaces between cellulose and hemicellulose, thereby providing support for lignocellulosic biomass. Lignin is known for its complex structure, and depending on where it was extracted, it has been found multiple unrepeatable structural units [40]. Certain characteristics are typical of most lignins; for example, lignins formed as a result of phenylpropanoid contain most of the methoxyl groups of wood, are resistant to acid hydrolysis and produce vanillin, syringaldehyde, and p hydroxybenzaldehyde when reacting with nitrobenzene in an alkaline solution [37].

As has already been mentioned, lignin comes in different forms and structures, and these can affect the properties and characteristics of its conversion into value-added products. These are usually classified according to the sulfur content thereof, as shown in Figure 6; the lignin with the lowest sulfur content is the most ecologically sound and has more applications due to the smell of other types of lignin [41]. Lignin that contains sulfur is derived from the paper industry and is divided into three categories: Kraft lignin, sulfite lignin, and hydrolyzed lignin. Lignin with no sulfur content is primarily created through biomass conversion technologies and solvent and soda pulping processes; this type of lignin has the highest number of applications in the creation of value-added products [41].

Multiple studies have been conducted to improve lignin extraction from lignocellulosic biomass, devise more effective extraction methods, and enhance the purity of extracted lignin, which employed several different methodologies. One of the greatest challenges in the lignin extraction process is the existence of a lignin–carbohydrate complex, which consists of different reactive components that are covalently bonded to hemicellulose; this is why it is necessary to extract lignin and achieve depolymerization in the most efficient manner possible to break these bonds, even though the reactivity of these components creates new C C bonds after depolymerization that form different intermediates with different properties [39].

There arephysicochemical technology and biological techniques that can be employed during lignin depolymerization: The first is achieved through the use of high temperatures and chemical additives, catalysts, and other compounds, which are commonly used in industrial areas. While the second technique, biological depolymerization, is a slower process, it is more environmentally friendly; this technique uses microorganisms such as bacteria and fungi to degrade lignin due to lignin-modifying enzymes that act by oxidative means, which can be classified according to their specific mechanism-of-action and according to the organism from which they originate [42].

There are also thermochemical techniques that can be employed. The most-used technique is the hydrogenolysis of lignin by means of solvents and catalyst systems. The most-commonly used solvents for this process are water, various alcohols, and compound solvents, which define the efficiency and distribution of products according to the specific characteristics of each compound, such as solubility, polarity, and the capacity of hydrogen supply [43]. There are also a variety of catalyst systems, including but not limited to acidic, basic, metallic, and ionic liquid catalyst systems. In the case of liquid acids, there is significant use of formic acid, sulfuric acid, hydrochloric acid, and phosphoric acid; the most commonly used solid acids are zeolite, metal oxides, metal phosphides, and Lewis acid; and basic catalysts include NaOH, KOH, MgO, and Na_2_CO_3_, and metallic catalysts such as Fe, Ni, Cu, and Co are also used [43]. Different systems have been devised that use solvent and catalyst variations, as well as different reaction times and temperatures; some examples of these are shown in Table 2.

The lignin extraction process consists of three steps, which are illustrated in Figure 7. The first step is the extraction of lignin from biomass, which can be achieved by several methods, including hydrolysis, oxidation or reduction fractionation, and a combination of pretreatment methods; followed by this. In the second step, the characterization process is continued by molecular weight determination, nuclear magnetic resonance analysis, and Fourier transform (FTIR). In the third and final step, the conversion process is conducted by different methods to obtain different products [39].

Regarding the environmental impact of this process, the use of products made from lignin have been shown to greatly assist efforts to combat climate change and the greenhouse effect, since one of the main by-products reduces the use of fossil fuels. Furthermore, the use of renewable, environmentally friendly biofuels derived from lignin to produce heat, electricity, and automobile fuel can further reduce the carbon footprint of these products and processes [41].

#### 2.5.2. Cellulose

Like lignin, cellulose is a biopolymer widely found in nature, mostly in plant bodies, some types of algae, the extracellular activities of microorganisms, and the structures of some marine animals. It has a semi-crystalline structure that is based on glycosidic bonds, which vary depending on the source from which the cellulose was obtained, resulting in multiple morphologies and structures that generate different physicochemical properties [56]. Cellulose is a biopolymer composed of groups of two or more repeated glucose units and these can form microfibrils when several cellulose strands are connected to one another. These microfibrils are typically found in the cell walls of trees and plants, and they provide flexibility, strength, and rigidity; the latter is because cellulose is hydrophilic, and absorbing water allows it to create a softer and more flexible wall [57].

The morphology of cellulose is presented in crystalline form, with a strong system of hydrogen bonds in intra and intermolecular forms, which generate a highly ordered, rigid, and strong crystalline structure. Four polymorphic structures of cellulose are currently known—named as I, II, III, and IV—and the mechanisms of each have been studied to understand the different structures of native cellulose, as shown in Figure 8 [58].

Cellulose is considered a compound with tremendous potential due to certain characteristics it possesses, such as its biodegradable, renewable, hydrophilicity, high absorbent potential, non-toxicity, and mechanical properties and its safety when discarded after use. The wide availability of cellulose in lignocellulosic materials such as wood biomass, coconut, mango, tomato, almonds, and pineapple has made it possible to generate several applications for crude and modified cellulose, with it being most commonly used as an absorbent to eliminate pollutants in wastewater and as a precursor for activated carbon [59]. Cellulose has also been found to have a property that facilitates the separation of solutions of water and oil; this was investigated, and several materials were created to treat oily wastewater in a simpler, more efficient, and environmentally friendly manner than traditional methods that caused other types of pollution and were typically more expensive [60].

Regarding the cellulose present in pineapple waste, more than 55% of the fruit is fibrous waste composed of lignin, hemicellulose, and cellulose, with cellulose representing a large proportion of this waste [61]; as was previously stated, the proportions of each vary according to the source of extraction, as is shown in Table 3 [62].

Due to its great abundance in pineapple residues, several treatments, and pretreatments for cellulose, microcellulose, and nanocellulose have been studied to identify applications in the field of biomaterials. Examples for this are biofuels such as bioethanol that are obtained by extracting cellulose from pineapple bagasse, which can be carried out through initial physical pretreatments of the waste such as drying and washing, followed by acid hydrolysis. After two phases of sulfuric acid for six hours, the fermentation of the obtained cellulose obtained is continued with the use of yeast such as S. cerevisiae, and the bioethanol is finally obtained with a final distillation [63].

The properties of nanocellulose include a high resistance to bending, which is helpful for the reinforcement of polymeric matrices and a great stabilizing potential for suspensions; this is why nanocellulose is used as an excipient in medications for enzymatic immobilization and to provide scaffolding for tissue engineering and biosensors. It is necessary to carry out two different acid hydrolysis processes on cellulose: HCl is added in the first step to obtain microcrystalline cellulose, which is then hydrolyzed with sulfuric acid, thereby reducing the lignin content, and allowing nanocellulose fragments to be obtained [64]. There are several methods to evaluate the treatments, properties, and composition of nanocellulose extracted from pineapple, which make it possible to obtain information of the chemistry, morphology, crystallinity, and thermal properties thereof that are useful to know when considering possible applications in value-added products. Some of the techniques used are high-resolution liquid chromatography, FTIR, scanning electron microscopy (SEM), X-ray diffraction analysis, and thermogravimetric analysis [65].

A study on nanocellulose demonstrated the great thermal stability thereof that makes it an affordable means to create bionanocomposites; it was also shown that cellulose pretreatments could be optimized based on tensile properties, reagent concentrations optimized in hydrolysis increases the amount of cellulose that can be obtained, and an efficient pineapple crushing process can improve mechanical properties of nanocellulose, such as its tensile strength [65]. In another study on the production of nanocellulose from pineapple leaves, acid hydrolysis was used as a pretreatment in conjunction with bleaching to eliminate other hemicellulose and lignin, after which a homogenization process that involved high shear and ultrasounds was carried out to obtain nanocellulose. After the final product was obtained, several studies were conducted, which determined that the degradation temperature of nanocellulose fibers is significantly higher than that of the initial fiber, which indicates that it has great thermal stability and commercial potential [66].

Nanocellulose is currently being investigated for a variety of valuable applications because of its potential for large-scale production, its mechanical properties, and its characteristics such as low thermal expansion, thermal stability, and biodegradability, among others. Some examples of applications include use in adhesives, paper-based materials, cement, food coatings, drilling liquids, transparent and flexible electronics, catalysis support structures, and biomedical materials [67].

#### 2.5.3. Hemicellulose

Hemicellulose is a mixture of several heteropolysaccharides in branched form, which varies depending on the location origin or extraction. The presence of pentoses, hexoses, and uronic acids stands out because of the amorphous nature of its branched shape, which is why hemicellulose is easier to solubilize and hydrolyze than cellulose and is, therefore, easier to obtain. There are two types of hemicelluloses: xylose, which is mainly found in hardwoods, and glucomannan, mainly found in softwoods. Hemicelluloses are found with lignin and cellulose in the cell walls of plants, where it acts as an interface between these compounds through hydrogen bridges, which allows them to fulfill several functions, such as the storage of substances and regulatory functions, providing structure, and controlling cell expansion [68].

One of the most common applications for all the components of lignocellulosic biomass is the creation of biofuels, such as bioethanol. A study on the role of hemicellulose in this process proved that it can be carried out by enzymatic hydrolysis with a two-phase multiscale process, and it was shown that after an optimal initial mixing that maximizes sugar yields and minimizes costs, said sugars are fermented until bioethanol is obtained with the use of yeasts or bacteria. This process has pretreatment, enzymatic hydrolysis, and fermentation [69]. Associated with this study, a different study indicated that pretreatment could improve the time and efficiency of enzymatic hydrolysis; microwave radiation was employed in conjunction with a hydrothermal-and-alkaline pretreatment, which resulted in a significant improvement in enzyme digestion, and a yield of 89.4% was obtained, compared to untreated residues wherein only 71.6% and 28.4% were obtained [70].

A study on hemicellulose in pineapple peels used an alkali-based method with 5%, 10%, and 15% concentrations and temperatures ranging from 36–65 °C, and a yield of more than 95% of the hemicellulose was obtained under the following conditions: a 15% concentration for 16 h at 45 °C. This study also discussed applying acid hydrolysis directly to the pineapple skin using diluted nitric acid such as xylitol. Finally, the study described the entire process of extracting hemicellulose and converting it into xylooligosaccharides and xylose, which are high-value end products that can be used to create biofuels and other products; this process is presented in Figure 9 [71].

Associated with this study, a different study affirmed that mass fractionation techniques are necessary to achieve the highest, most efficient recovery of hemicellulose, of which a hydrothermal pretreatment was shown to be the most prominent and valuable due to the use of high temperatures and pressures; this study also discussed several pretreatments of this type, such as supercritical water hydrolysis and subcritical water hydrolysis, as shown in Figure 10; these make it possible to recover a large number of products and result in several applications related to bioresins, bioenergy, hydrogels, bioethanol, food industry, among others [72].

### 2.6. Medical Applications

Finally, the functions, varieties, and characteristics of several biomedical applications that are manufactured from by-products obtained from the biorefinery of pineapple waste will be described.

#### 2.6.1. Hydrogels Force Cellulose Sources

Hydrogels are products that can be used in the field of biomedicine, which form three-dimensional (3D) cross-linked polymeric networks with properties that allow for the absorption, swelling, and retention of large amounts of water without dissolving [73]. These properties have allowed application of this product in the area of biomaterials for cell cultures, tissue engineering, drug administration, and wound dressing [74]. Hydrogels are also considered to be very attractive products due to their advantage in relation to other materials of being biocompatible, biodegradable, and low in toxicity [75]. An example of this is the hydrogel derived from lignin created by Xu et al. (2021), which was manufactured through the free radical polymerization of sulfobetaine methacrylate (SBMA) and partially methacrylated lignin, resulting in a functional cytotoxic hydrogel with antifouling and antimicrobial properties because lignin serves as a hydrogel skeleton and antibacterial agent; while the high hydration of the SBMA reduces the adsorption of proteins and prevents the adhesion of bacteria [76].

Hemicelluloses are also known to have properties that allow them to be transformed into hydrogels through chemical, physical, and enzymatic crosslinking methods; the enzymatic method is a more-recent alternative to hydrogels because it does not use toxic chemicals [77]. An example of this type of hydrogel was created by Liu in 2019 from waste hemicelluloses; polyvinyl alcohol was added and integrated into the hemicelluloses and resulting in improved swelling properties; this hydrogel was described by means of SEM, FTIR, and TG techniques to determine such aspects as morphologies, structure, and thermal stability [78].

As it relates to cellulose, bacterial nanocellulose that is free of hemicellulose, lignin, and pectin is known for its notable biocompatibility and its water retention capabilities, flexibility, mechanical properties, and hydroxyl groups similar to those of native tissues. For this reason, a significant amount of research has been conducted on its application in the area of tissue engineering, specifically in the creation of hydrogels to aid in the regeneration of hard tissues. This type of hydrogel was first used in the 1980s to treat severe burns, skin grafts, and chronic skin ulcers, and it started to be used to create artificial blood vessels in microsurgery at the beginning of the 21st century [79]. To date, extensive research has been conducted on the use of this type of hydrogel for the formation of 3D scaffolds for bone and dental structures due to its notable structural stability, traction resistance, and capacity for cell adhesion and the proliferation of fibroblast cells, and human osteoblastic cells. Some of the studies that have been conducted to investigate the use of biomaterials for tissue regeneration are delineated below in Table 4.

In addition to these studies, a different study carried out two approaches—one pre-surprise and the other during in situ adsorption—to investigate the application of hemicelluloses (i.e., galactoglucomannan, xyloglucan[XG], and xylan) as crosslinking agents, which adjust the cellular behavioral properties in nanofibrillated cellulose hydrogels (NFC) during wound healing. In the pre-surprise method, XG was found to be the hemicellulose with the greatest adsorption capacity in NFC, and it also demonstrated the highest efficacy when supporting cell growth and proliferation of the cell, making it a highly functional polysaccharide that can be applied in wound healing [87].

#### 2.6.2. Enzymatic Products

Bromelain is a proteolytic enzyme found in pineapple residues; while it is mostly found in the stem, it can also be obtained from the pulp, core, and peel. Obtaining this enzyme involves a 10-step process—reception, washing, cutting, grinding, pressing, tangential microfiltration, ultrafiltration, microencapsulation, drying, and mixing—which yields a yellow powder [88].

Significant research on the properties of bromelain has been conducted. This enzyme was found to be an anti-inflammatory agent that reduces prostaglandin, and it can also be used as an analgesic agent to treat muscle injuries, reduce inflammation from arthritis, and ease the pain of surgical cuts in the perineum. Studies have also shown that bromelain acts as an anticancer and antimicrobial agent; in fact, this enzyme was found to inhibit the first phases of the metastatic process [89]. As an example, a study was conducted that demonstrated that bromelain inhibited cell growth in tumor cell lines [90]; this also occurred in a study on gastric carcinoma cell lines, in which bromelain therapy caused to decrease significantly; and the inhibition of MCF-7 breast cancer cells observed after an oral bromelain treatment [91].

Another well-known application of bromelain is in the treatment of burns and wounds; this enzyme in this capacity is considered an alternative treatment to surgery because it collaborates in the elimination of necrotic tissues and accelerates the skin’s healing process. As a result, there are currently medicines based on proteolytic enzymes such as bromelain available as either a powder or a gel, which is applied to the skin for 24 h after a significant burn has occurred [92].

Finally, a study on the collagenase activity found in bromelain was conducted; here, the enzyme was immobilized with the use of a gold nanoparticle interface, which resulted in improvements in the thermal stability and enzymatic activity of this biomaterial in therapeutic applications that rely on the anti-inflammatory, anticoagulant, antitumor, antimetastatic in vitro properties and in vivo immunogenicity properties [93].

#### 2.6.3. Drug Delivery

The application of these compounds as materials to deliver drugs throughout the body has been extensively researched. Several studies have investigated the delivery of different types of drugs concerning lignin, cellulose, and hemicellulose; some of these studies are presented below.

In a study on the synthesis of lignin-based polymeric nanoparticles for use in the delivery and administration of drugs during a cancer treatment, was conducted because of problems posed by these types of drugs, specifically minimal water solubility, rapid elimination from circulation, poor targeting of tumor cells, and inadequate tissue penetration. The synthesis process of this material, which is summarized below in Figure 11, resulted in a conjugated folic acid-polyethylene glycol-alkaline lignin base and self-assembly. The material that administered the anticancer drugs was successfully and efficiently obtained; it also demonstrated high loading efficiency, robust stability, and good biocompatibility, which in turn resulted in improved blood circulation and cell uptake, and the conclusion of this study was that this material was capable of significantly inhibiting advanced tumor growth [94].

Another study synthesized lignin-based pH-sensitive nanocapsules for the controlled delivery of hydrophobic molecules, which can be found in certain medications, essential oils, and antioxidants. This synthesis was conducted by a three-step interfacial miniemulsion polymerization: The first step involved etherification to graft the lignin with allyl groups in an emulsion system (i.e., water-oil), followed by a reaction of the lignin with a cross-linking agent to form nanocapsules. Finally, the success and efficiency of this polymerization were determined and characterized by FTIR and TEM tests with a linear release at pH 7.4 and in accordance with the Korsmeyer–Peppas model at pH 4 [95].

Studies have also shown that fractionated Kraft lignin can be easily used in the creation of drug microcapsules of various shapes, such as a hemisphere, a bowl, mini-tablets, or spheres with a single hole; this is due to the physicochemical properties of lignin in the water-oil interface. An emulsion solvent evaporation protocol was utilized in this study, and it was determined that the employed method allowed for the creation of a high number of capsules that could be used in a wide range of medical applications in different drugs [96].

Lignin has been used to create biomaterials that can help treat or combat oxidative stress in compounds containing poly (L-lactic acid). It was achieved through a ring-opening polymerization with different alkylated lignin contents that ranged from 10–50%; after which the lignin was combined with poly(L-lactide) to manufacture nanofibrous compounds through electrospinning, which efficiently eliminated radicals and resulted in good compatibility, all of which make this a product that can be widely used in the field of biomedical materials [97]. Figure 12 below depicts the synthesis route used in this study on lignin and PLA biopolymers.

It should be noted that lignin is not the only component used in this area. Studies have also been conducted on the potential of cellulose and nanocellulose for drug delivery. These were based on the fact that nanocellulose crystals are nontoxic and demonstrate undirected absorption, which suggests that they may have great potential as drug carriers for some specific deliveries [98]. The research was conducted to investigate the application of cellulose polymers in the delivery of mucoadhesive nasal medications; in this study, the properties of methylcellulose, hydroxypropylmethylcellulose, sodium carboxymethylcellulose, and cationic hydroxyethylcellulose (cationic-HEC) polymers, such as the rheology, ciliary beat frequency, and permeability throughout the nasal tissue, were evaluated, and all of the polymers presented unique advantages that allowed for the improved administrations of nasal drugs. This study concluded that even though the cationic-HEC was better able to help acyclovir penetrate nasal mucosa, none of the polymer characteristics studied stood out, and more cellulose derivatives needed to be developed for this type of drug [99].

#### 2.6.4. Antimicrobial Products

Antimicrobial compounds obtained from pineapple can be encapsulated, immobilized, or added to hydrogels to improve medical devices’ antimicrobial properties. The most studied antimicrobial compounds in *Ananas comosus* are bromelain and saponin. Their mechanisms of action are associated with the disruption of the bacterial cell, producing the cell wall to weaken. Bromelain has antimicrobial properties against Gram-positive and Gram-negative bacteria [100]. Saponins interact with cholesterol and proteins, creating holes that cause cell permeability [100,101]. Pineapple ethanolic extract contains antimicrobial compounds that can be used to produce antimicrobial formulations. Some compounds are thiamine, niacin, riboflavin, p-coumaric acid, caffeic acid, ferulic acid, sinapic acid, quercetin, ananasic acid, subaphyllin, beta carotene, lutein, cyamin, cyaniding 3,5,3′-triglucoside, beta-sitosterol, campesterol, and hydroxysitosterol [102].

Biogenic silica rosettes (BSR) are another sub-product of the nanocellulose extraction process from pineapple peels that present antimicrobial properties [64]. These rosettes have ten micrometers in length, 5 µm in height, and are formed by 300 nm granules. The minimal inhibition concentration of these BSR is lower than similar length non-nanostructured silica. Castro et al. reported that the monolayer of these structures on the polydimethylsiloxane surface disrupted the bacteria membrane, decreasing the bacteria adhesion and growth [103]. These complex microstructures, formed mainly by Si, O, and C, can be used to produce antimicrobial coatings on different polymeric materials for medical devices.

### 2.7. Modeling and Simulation Studies of Pineapple Derivates

The properties of nanocellulose and aqueous hydrogels have been investigated using molecular modeling and simulations. The reported literature focuses on applications such as hydrogels for biomedical purposes, drug delivery emulsion formation for different formulations, and pollutants absorbent materials.

Trentin et al. investigated the relative hydrophilicity of cellulose surfaces using molecular dynamics simulations. The water molecules fully spread onto highly hydrophilic, such as Iα (010), Iα (110), Iβ (010), and Iβ (110) faces, among others. They showed that surface oxidation is a key factor influencing surface hydrophilicity nonlinearly [104]. Mehandzhiyski et al. reported the Young Modulus value of 110 GPa of one nanocellulose fibril; The calculated diffusion coefficients of nanocellulose show an exponential decay with the increase of the length and vary between 17 and 1 × 10^−12^ m^2^/s for lengths between 50 and 1200 nm, respectively [105]. Dong et al. studied the nanocellulose thermal properties using molecular dynamics. They reported that they depend on the nanocellulose length and cross-sectional area [106]. The cellulose hydrogel mechanical properties were studied using a poroelastic mechanical model, suggesting that mechanical response depends on the mechanical properties of the cellulose network, deformation time, and the water movement within the structure [107].

Lombardo et al. focused on the drug delivery capability of nanocellulose, demonstrating that poorly soluble drug molecule furosemide was spontaneously adsorbed on cellulose nanofibers, and its adsorption was driven by charge neutralization between positive nanocellulose fibers, and the negative furosemide surface charge, Figure 13 [108]. Molecular dynamics simulations were carried out by Ning et al. to gain insights into the interactions between nanocellulose ((110), (100), (1–10), (010)) crystal planes and collagen; showing that the structure of collagen was preserved during binding and suggested that this structural preservation is correlated with the good biocompatibility [109]. Lee at al. investigated via molecular dynamics simulation the cellulose properties as a Pickering emulsion stabilizer [110]. The interaction of composites formed by poly (2-hydroxyethyl methacrylate) (PHEMA), poly (methacroylcholine chloride) (PMACC) or poly (methacroylcholine hydroxide) (PMACH) and bacteria nanocellulose were characterized by density functional theory, showing that the matrix and filler interact forming a homogeneous material and not mixtures of totally independent domains [111]. Finally, the Fe_3_O_4_-nanocellulose has been studied by density functional theory as a material to remove mercury ion pollution [112].

The surface modification has been reported by Chen et al. using atomistic molecular dynamics. They reported that the modification with acetyl groups could disrupt the near-crystalline order at the interface between two aggregated cellulose nanoparticles [113]. Another product obtained from pineapple that has been modeled is bromelain. The reported studies have focused on elucidating the secondary and tertiary structure, intermolecular and intramolecular hydrogen bonds, structure stability of the enzyme, and temperature and osmolytes’ effect on its denaturalization [114,115]

## 3. Conclusions

As shown throughout this work, pineapple waste is highly useful because it allows different types of by-products to be obtained through biorefinery, specifically cellulose, hemicellulose, lignin, and enzymes with a high added value. In particular, the proteolytic enzyme bromelain is commonly used as an anti-inflammatory agent in many commercial products. Moreover, scientists have also discovered its potential in the treatment of burns, as antimicrobial agent, and even as inhibitor of cell growth in tumor cell lines. Other by-products, like cellulose, hemicellulose, and lignin, although are not obtained only from pineapple waste, have an undoubtedly huge commercial application.

Applying the circular economy concept to the pineapple production-and-processing industry addresses and contributes to solve waste-related environmental issues and creates an opportunity to build a high-profit market that focuses on manufacturing products with biomedical applications. Pineapple processing industry of different food products generates a high amount of waste which is easily accessible and is usually concentrated in a single point, which reduces the cost of retrieving it for additional treatments. Moreover, a treatment plant could be built next to processing facilities, which would offer the opportunity to develop a closed-loop processing system.

Pineapple-producing countries, most of which are located in Central and South America, have a unique opportunity to generate export markets that would allow them to recover from the economic crisis produced by the COVID-19 pandemic. Furthermore, thanks to the reuse of waste, these new markets would lead to a sustainable economy aligned with world trends toward a circular economy. State policies that allow the development of these technologies are critical for their implementation. In this regard, Costa Rica has demonstrated its interest in positioning itself as a leader in the field of biorefinery due to proposals that would facilitate the full use of all residual biomasses; the country’s overwhelming support for research on this issue and the implementation of policies at the local level to promote the development of new productive activities based on the full use and valorization of residual biomass is evidence of this interest.

## Figures and Tables

**Figure 1 polymers-14-04864-f001:**
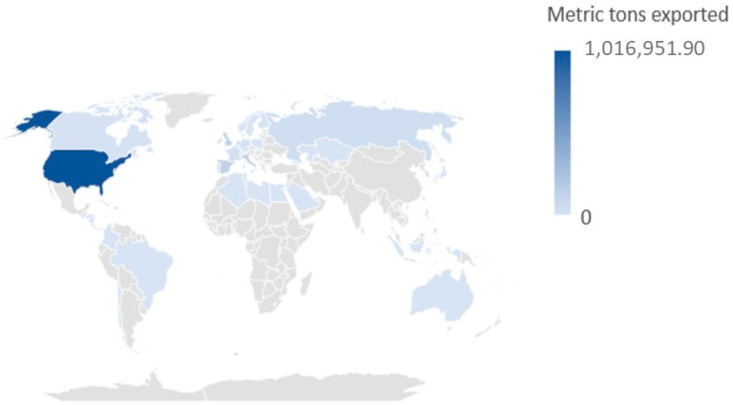
Geographical distribution of the Costa Rican pineapple exportations during 2020 (the first year of the COVID-19 pandemic).

**Figure 2 polymers-14-04864-f002:**
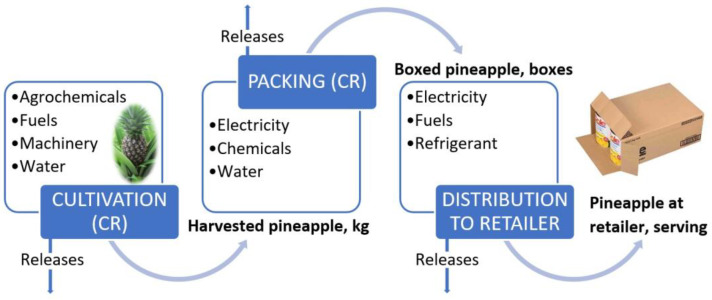
Diagram of fresh pineapple system [5].

**Figure 3 polymers-14-04864-f003:**
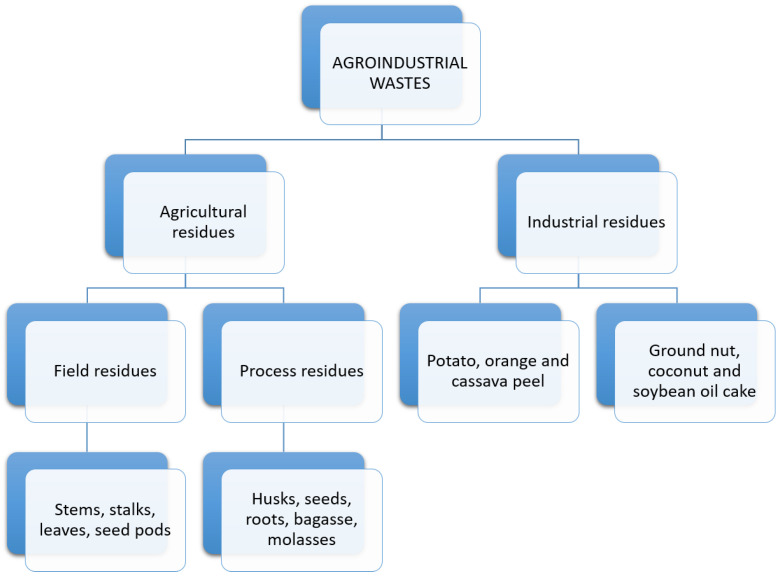
Agro-industrial wastes and their types [7].

**Figure 4 polymers-14-04864-f004:**
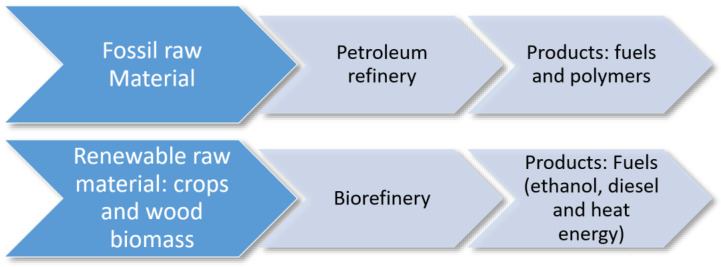
Comparison between biorefinery and oil refinery [27].

**Figure 5 polymers-14-04864-f005:**
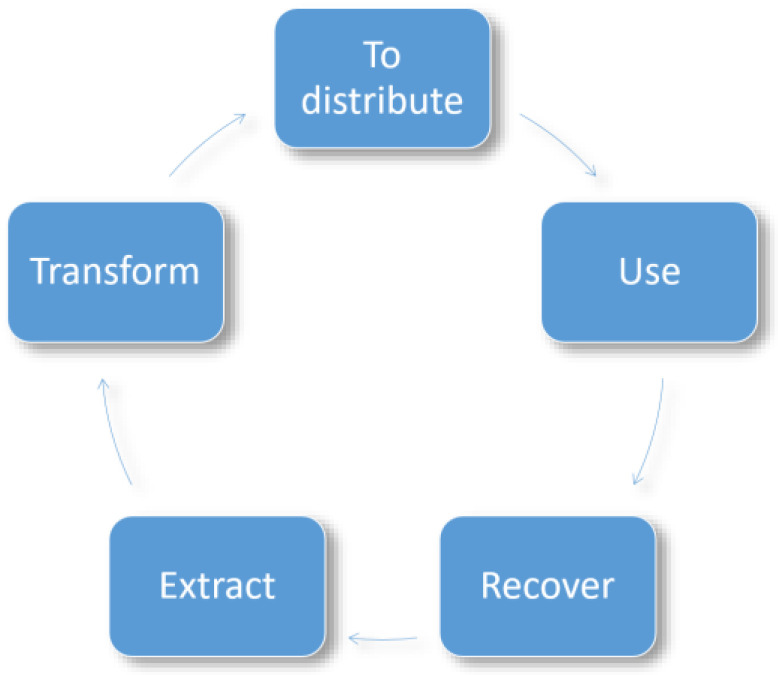
Fields of actions in the circular economy [34].

**Figure 6 polymers-14-04864-f006:**
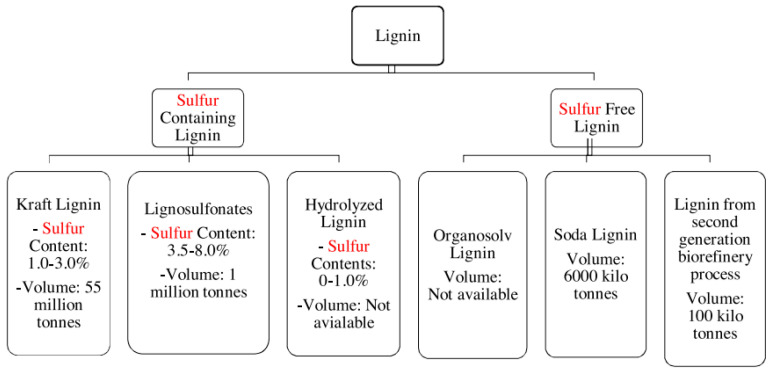
Different sources of lignin and their current volume on scale global [41].

**Figure 7 polymers-14-04864-f007:**
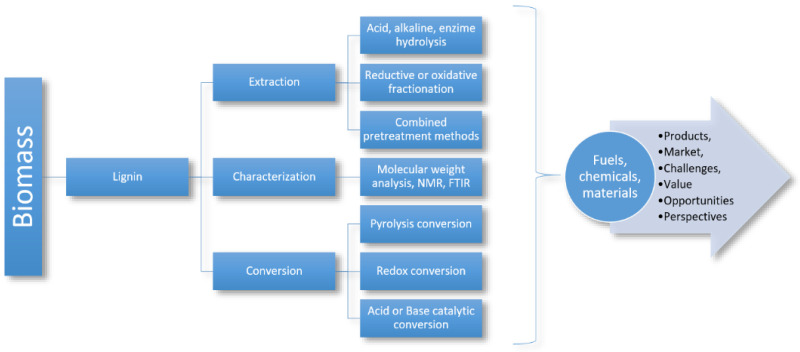
Overview of valorization of lignin to valuable products [39].

**Figure 8 polymers-14-04864-f008:**
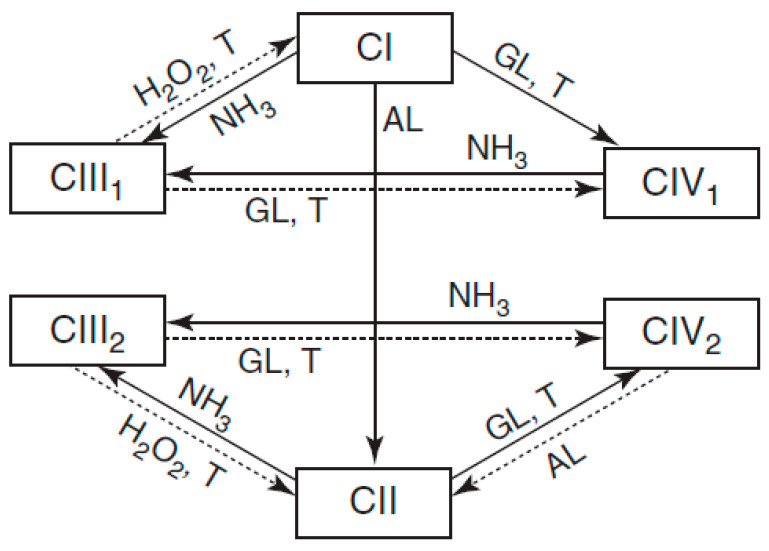
Scheme of transitions between various polymorphic crystalline forms of cellulose CI (native cellulose), CII (cellulose), CIII1 and CIII2 (cellulose III1 and III2), and CIV1 and CIV2 (cellulose IV1 and IV2) [58].

**Figure 9 polymers-14-04864-f009:**
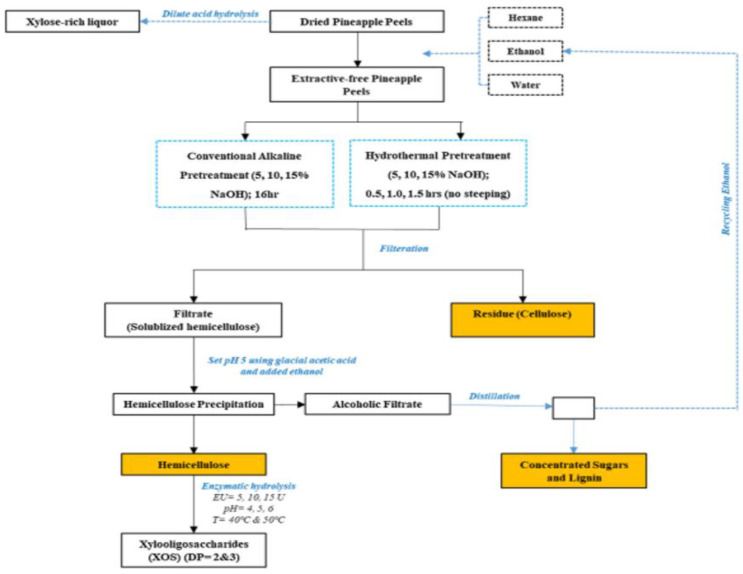
Schematic for extraction of hemicellulose from pineapple peel waste and its valorization into xylooligosaccharides [71].

**Figure 10 polymers-14-04864-f010:**
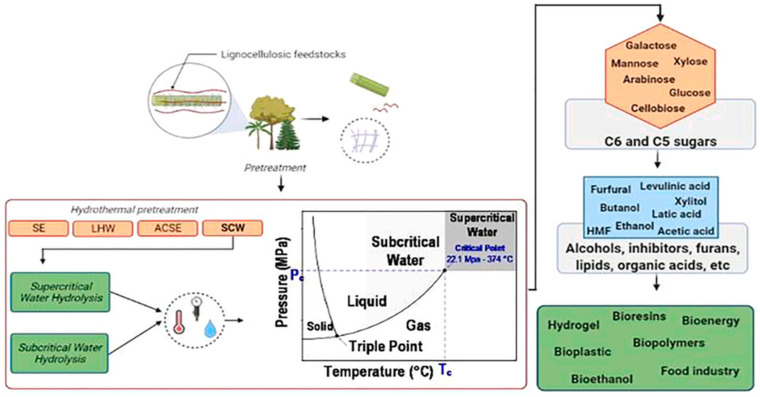
Hydrothermal pretreatments applied to lignocellulosic biomass and the possibility of multi-product recovery [72].

**Figure 11 polymers-14-04864-f011:**
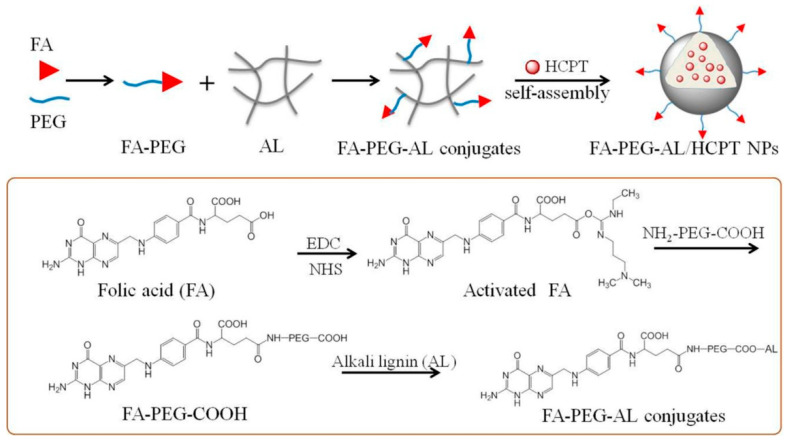
Synthesis of FA-PEG-AL/HCPT NP [94].

**Figure 12 polymers-14-04864-f012:**
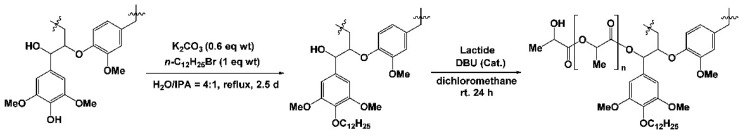
Synthesis route used for lignin and PLA biopolymers [97].

**Figure 13 polymers-14-04864-f013:**
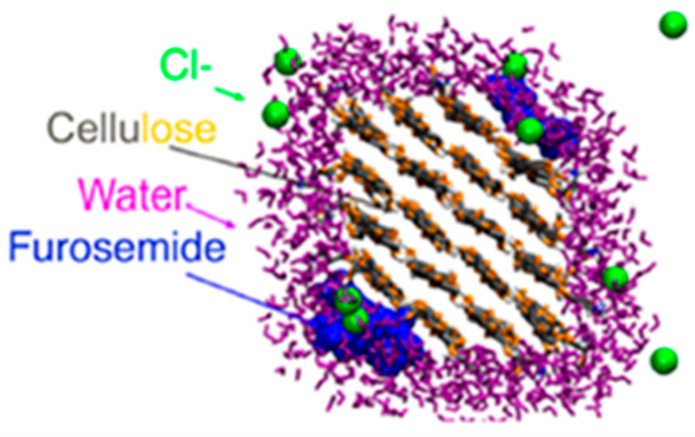
Snapshot of the sorption simulation of furosemide on a charged cellulose nanofiber [108].

**Table 1 polymers-14-04864-t001:** Details of the Costa Rican pineapple exportations during 2020 (first year of COVID-19 pandemic outbreak). Source: own design based on the Costa Rican Foreign Trade Promoter (PROCOMER) data [4].

Destination	Metric Tons	Percentage
USA	1,016,951.89	51.01
European Union	715,785.82	35.90
United Kingdom	142,386.54	7.14
Russia	45,673.77	2.29
Turkey	28,414.61	1.43
Other countries	44,554.93	2.23
Total	1,993,767.56	100.00

**Table 2 polymers-14-04864-t002:** Conversion of lignin in different solvents [43].

Feed	Solvent	Catalyst	Reaction Conditions		Major Product	Major Product Yield (%)	Ref.
			Temp. (°C)	Time (h)			
Poplar wood sawdust	Alkaline aqueous solution	NiAl alloy	220	3	Aromatic monomers	76	[44]
Kraft lignin	Alkaline water	Ni/ZSM-5 zeolite	200	4	Monomers	17	[45]
Birch sawdust	water	Pd1Ni4/MIL-100(Fe)	130–180	6	Phenol + acetophenone	56.3	[46]
Organosolv lignin	Methanol	Cu20PMO	300	24	Liquid phase products	26.3	[47]
Hydrolyzed lignin	Methanol	PD/C, CrCl3	300	4	Monomers	60–86	[48]
Asian lignin	Ethanol	Pt/C	350	2/3	Lignin-oil	77.4	[49]
Kraft lignin	Isopropanol	Ni-Cu/H-Beta	330	3	cycloalkanes	40.39	[50]
beech dioxasolv lignin	Methanol	W/C	200	6	Lignin. Oil	56.4	[51]
Birch lignin	Aqueous solution	Ru/Nb_2_O_5_	250	20	Liquid hydrocarbons	35.5	[52]
Enzymatic lignin	Water	Ru/Nb_2_O_5_	250	20	Hydrocarbons	99.6	[53]
Alkali lignin	Ethanol	Ni_2_P/SiO_2_ + CuMgAl_2_O_3_	340	4	Monomers	53	[54]
Corn stover lignin	N-octane	Ru/Al_2_O_3_ + Hf (OTf)	250	4	Hydrocarbons	>30	[55]

**Table 3 polymers-14-04864-t003:** Lignocellulosic material composition [62].

Lignocellulosic Material	Cellulose (%)	Hemicellulose (%)	Lignin (%)
Pineapple crown (leaves)	45.53 ± 1.17	21.88 ± 0.22	13.88 ± 1.70
Pineapple peel	40.55 ± 1.02	28.69 ± 0.35	10.01 ± 0.38
Pineapple core	24.53 ± 1.68	28.53 ± 1.37	5.78 ± 0.429
Pineapple stubble	32.2	21	2.83

**Table 4 polymers-14-04864-t004:** Examples of the potential of nanocellulose biomaterials in the regeneration of bone tissues [79].

Composition	Scaffold Form	Cell/Drug/Biomolecule	Features	Ref
BNC	Membrane	NIH-3T3 fibroblast cells	Suitable biocompatibility and enhanced cell viability, remarkable formation of large new bone area	[80]
BNC	Membrane	-	Low biocompatibility and large amount of mature connective tissue in filling the defect (adult male rat)	[81]
BNC	Nanofibrous	BMP-2, C2C12 cells	Suitable biocompatibility and osteogenic differentiation of fibroblast-like cells, and BNC scaffold with BMP-2 exhibited more bone formation and higher calcium content than that of BNC only	[82]
BNC/collagen	Fibrous	UCB-MSCs and NIH3T3 cells, BMP-2, dexamethasone	Favorable cell adhesion and growth, more up-regulated osteogenic markers and remarkably uplifted proteins and calcium deposition, and positive signals (-smooth muscle actin) for neovascularization	[83]
BNC/Gel	Nanofibrous	NIH-3T3 fibroblast cells	Decreased crystallinity and improved thermal stability, Enhanced Young’s modulus and decreased tensile strength, and excellent biocompatibility	[84]
BNC/fisetin	Nanofibrous	BMSCs	Decreased crystallinity and improved thermal stability, Enhanced Young’s modulus and decreased tensile strength, and excellent biocompatibility	[85]
BNC/HAp	Nanofibrous	-	3D porous network with homogenous precipitation of carbonated-HAp crystals on BC fibers	[86]

## Data Availability

Not applicable.

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
