# Peer review of "Pineapple Agro-Industrial Biomass to Produce Biomedical Applications in a Circular Economy Context in Costa Rica"

_polymers, 2022, doi:10.3390/polym14224864_

Round 1
Reviewer 1 Report
The authors submitted "Pineapple agro-industrial biomass to produce biomedical applications in a circular economy context in Costa Rica" to publish in the Polymers. The quality of this manuscript needs further modification to be published in the Polymers. My comments are listed herby:
1. The subject of the review is interesting, but the current Abstract doesn’t exactly reflect the title and content of the manuscript.
2. The review is not to write the history of the development of the discipline, but to collect the latest information, obtain the latest content, and deliver the latest information and scientific research trends to the readers promptly.
3. One of drawbacks in the manuscript is that the evaluation is limited. It is likely difficult to obtain good benchmarks for this type of evaluation. Ideally, this type of work (post-processing of pineapples) should be evaluated over many problem instances.
Author Response
Reviewer 1:
The authors submitted "Pineapple agro-industrial biomass to produce biomedical applications in a circular economy context in Costa Rica" to publish in the Polymers. The quality of this manuscript needs further modification to be published in the Polymers. My comments are listed herby:
- The subject of the review is interesting, but the current Abstract doesn’t exactly reflect the title and content of the manuscript.
- The review is not to write the history of the development of the discipline, but to collect the latest information, obtain the latest content, and deliver the latest information and scientific research trends to the readers promptly.
- One of drawbacks in the manuscript is that the evaluation is limited. It is likely difficult to obtain good benchmarks for this type of evaluation. Ideally, this type of work (post-processing of pineapples) should be evaluated over many problem instances.
- The subject of the review is interesting, but the current Abstract doesn’t exactly reflect the title and content of the manuscript.
It was changed:
Pineapple is a highly demanded fruit in international markets due to its unique appearance and flavor, high fiber content, vitamins, folic acid, and minerals. It makes pineapple production and processing a significant source of income for producing countries, such as Costa Rica. This review collects bibliographic information dating back to the beginnings of pineapple production in Costa Rica to the state of the market today. It details the impacts of its production chain and proposes biorefinery as a solution to environmental problems. Besides the potentiality of new sustainable markets to contribute to the post-COVID-19 economy in Costa Rica is highlighted. The general characteristics of pineapple by-products -cellulose, hemicellulose, lignin, and other high-value products like bromelain y saponin- are described, as well as the primary processes for their ex-traction via biorefinery and main applications in the medical field. Finally, a brief description of the main works in the literature involving modeling and simulation studies of pineapple by-products properties is included.
- The review is not to write the history of the development of the discipline, but to collect the latest information, obtain the latest content, and deliver the latest information and scientific research trends to the readers promptly.
Some details involving more recent scientific information were changed. This is reflected in the new abstract. Document information was also handled more appropriately. It is important to emphasize that for Costa Rica, it is important to make its biorefinery work known to the world.
- One of drawbacks in the manuscript is that the evaluation is limited. It is likely difficult to obtain good benchmarks for this type of evaluation. Ideally, this type of work (post-processing of pineapples) should be evaluated over many problem instances.
Yes, you are right. It is likely difficult to obtain adequate benchmarks. For example, in 2018 we report a new material from pineapple wastes: Corrales-Ureña, Y.R., Villalobos-Bermúdez, C., Pereira, R. et al. Biogenic silica-based microparticles obtained as a sub-product of the nanocellulose extraction process from pineapple peels. Sci Rep 8, 10417 (2018). https://doi.org/10.1038/s41598-018-28444-4
It was a good report… easier compared with this…

Reviewer 2 Report
Overall, the topic of the manuscript is of considerable relevance to the field of high-value products created from agricultural residues. In order for this review to be valuable to researchers in the respective field, I would suggest to shorten the general information on lignocellulosic residues and their treatment, while focussing more on the specific properties and applications of pineapple residues. Further comments on content and language are included in the attached file.

Author Response
Reviewer 2
Overall, the topic of the manuscript is of considerable relevance to the field of high-value products created from agricultural residues. In order for this review to be valuable to researchers in the respective field, I would suggest to shorten the general information on lignocellulosic residues and their treatment, while focussing more on the specific properties and applications of pineapple residues. Further comments on content and language are included in the attached file.
- Please rephrase to make clear that you are reporting on the problem of environmental pollution due to the inexisting / inadequate treatment of residues from pineapple production.
Abstract was changed:
Pineapple is a highly demanded fruit in international markets due to its unique appearance and flavor, high fiber content, vitamins, folic acid, and minerals. It makes pineapple production and processing a significant source of income for producing countries, such as Costa Rica. This review collects bibliographic information dating back to the beginnings of pineapple production in Costa Rica to the state of the market today. It details the impacts of its production chain and proposes biorefinery as a solution to environmental problems. Besides the potentiality of new sustainable markets to contribute to the post-COVID-19 economy in Costa Rica is highlighted. The general characteristics of pineapple by-products -cellulose, hemicellulose, lignin, and other high-value products like bromelain y saponin- are described, as well as the primary processes for their ex-traction via biorefinery and main applications in the medical field. Finally, a brief description of the main works in the literature involving modeling and simulation studies of pineapple by-products properties is included.
- Had: …. modifications had to:
- Such as the creation of: deleted
- Have needed: deleted
- g. used
- Explain how the dependability of Costa Rica on pineapple exports could be alleviated by the concept of a bioeconomy!: Lignocellulosic materials can produce other materials with high added value, such as crystalline nanocellulose, and dextrans, for clinical uses, as well as energy production, among others. Likewise, the pineapple circular economy can be promoted and even con-tribute to creating companies that contribute to the country's economy.
- Aims: changed
- Provide at least a brief description of how you proceeded to access the relevant bibliographical information: from Costa Rican institutions such as the National Chamber of Pineapple Producers and Exporters - CANAPEP and the Ministry of Agriculture and Livestock - MAG,
- Avoid repetitions!: There is a high demand for this fruit in the international market, mostly due to its flavor. Its high fiber content and potassium, iodine, carbohydrates, fiber, and vitamins A, B, and C, have prompted recommendations to consume pineapple to treat such diseases as fluid retention due to its diuretic effects, hypertension, cholesterol, anemia, poisoning due to its purifying action, immune system disorders due to its ability to strengthen defenses and assist in the formation of both red and white blood cells, cardiovascular problems, and obesity, among others [10, 11].
- From: deleted… “due to the increase in Philippines exports and importations into China.”.
- Deleted commoly used
- Deleted jobs
- Shorten this paragraph: Most Costa Rican pineapple exports are sent to the United States and the European market. Costa Rica also exports to the Central American, Middle Eastern, and Asian sec-tors [1]. Also, the pineapple was exported to China for the first time in 2017, which re-sulted in increased business on the Asian continent [17].
The main destinations detail of Costa Rican pineapple exportation in the first year of the pandemic outbreak are shown in figure 1. The USA is the most important market, with 51% of the total exports. The European Union is the second most important market, with almost 36% of the total distributed in several countries. As combined consumers, the United Kingdom, Russia, and Turkey represent more than 10% of Costa Rican pineapple exportations. Finally, the rest of the countries represent 2,23% of the exports.
- This information appears implausible and unclear. Where do the remaining 98.5 % of solid waste come from? It is estimated that more than 4,000 tons of solid waste are produced each day in Costa Rica. In general, this waste has been poorly handled; only 78% is collected by the municipalities and taken to sanitary landfills, and the location of the remainder of the waste and the environmental impact thereof is unknown [18].
- Under which conditions? (in anaerobic conditions)
- Aqueducts are not bodies of water. Do you mean aquifers? Yes, it means aquifers, it was changed.
Seeds:
- The figure is helpful in principle, yet it would be more informative if you complemented it with the specific wastes from pineapple production: …. “In general, the pineapple plant is composed of eight parts, the names of which correspond to the whole plant—the stubble, crown, shell, flesh, stem, roots, leaves, and heart—and all create a significant amount of lignocellulosic biomass (i.e., lignin, cellulose and hemicellulose); however, as is shown in Figure 4, most of this biomass is discarded after the pineapple, and the by-products mentioned above are produced [26]…”.
- Refer here to amended Figure 4!. It was used: as is shown in Figure 4 in: In general, the pineapple plant is composed of eight parts, the names of which correspond to the whole plant—the stubble, crown, shell, flesh, stem, roots, leaves, and heart—and all create a significant amount of lignocellulosic biomass (i.e., lignin, cellulose and hemicellulose); however, as is shown in Figure 4, most of this biomass is discarded after the pineapple, and the by-products mentioned above are produced [26].
- If you refer here to energy generation, this definition is much too narrow. Also, "first round of processing" remains unclear unless you explain the general concept of biorefineries, which is typically composed of the following four components: Feedstock/raw materials; platforms; processes; products. After having outlined the general concept, you can proceed to explaining the category of "lignocellulosic biorefinery". General concept of biorefineries is typically composed of feedstock/raw materials; platforms; processes; and products. In this case, using pineapple wastes to create val-ue-added products, by-products, and energy.
- Introduce "Lanotec"!: by National Nanotechnology Laboratory Lanotec in Costa Rica.
- As you were talking of "biorefinery" and "bioeconomy", the jump to "nanotechnology" appears very abrupt and incomprehensible to me: and nanomaterials (e.g. nanocellulose).
- No, certainly not in the earth's crust! I surmise you mean in the biosphere: deleted.
- The second part of the sentence has nothing to do with the first part. Separate information on the general chemistry of lignin from that on its role within the focus of your manuscript, i.e. residues and waste from pineapple production!: From pineapple peel wastes, lignin is in the range of 15– 30% by dry weight [42].
- On a global scale? Yes, on a global scale.
- step is the extraction of lignin from biomass, which can be achieved by several 402 methods, including hydrolysis, acidic, alkaline hydrolysis, enzymatic hydrolysis, oxidation: deleted.
- Clarify abbreviation!: Fourier transform (FTIR):
- Legend is missing!: Completed: Figure 9. Scheme of transitions between various polymorphic crystalline forms of cellulose CI (native cellulose), CII (cellulose), CIII1 and CIII2 (cellulose III1 and III2), and CIV1 and CIV2 (cellulose IV1 and IV2) [61].
- Examples for this: changed.
- Could deleted.
- Irrelevant in this functional context!:
- Structural: deleted.
- What is Chapter 3 then?:
- What changed.
- A large number of your references are in Spanish language. For a paper in an English journal, this is not acceptable, unless relevant literature sources are not available in English language. This is certainly not the case for key topics such as biorefineries or circular economy, for which fundamental references were published in English. Also, in the introductory section of the manuscript you are citing various internet resources without providing sufficient bibliographical information. Yes, You are right. It is not possible to look for information in English from government information.
- Reference 15 was
- Reference 25 was completed.

Round 2
Reviewer 1 Report
My questions are well addressed, and it can be published.
Author Response
No comments
Reviewer 2 Report
Dear authors
Unfortunately, my comments regarding the large number of inappropriate references from the WWW and in Spanish language were mostly ignored.
Please, use decimal points in Tables 2 and 3!
Author Response
Part 2.
Reviewer 2:
- Unfortunately, my comments regarding the large number of inappropriate references from the WWW and in Spanish language were mostly ignored.
This article is focused in the Costa Rican context, due to this reason, we need to present an extensive revision of references that are not available in English, this www. references belong to the national and agricultural institutions that will not be available in other format or language.
At this regard, we believe that this fact is extremely beneficial for the quality of this review, because it allows us to extract official information, which was originally presented in the Spanish language, and now we make this same information available in the English language through this manuscript, to a broader audience.
Thanks a lot for your kind recommendation
- Please, use decimal points in Tables 1, 2 and 3! We changed them.
Table 1. Details of the Costa Rican pineapple exportations during 2020 (first year of Covid-19 pandemic outbreak). Source: own design based on the Costa Rican Foreign Trade Promoter (PROCOMER) data. [4].
Destination |
Metric tons |
Percentage |
USA |
1016951.89 |
51.01 |
European Union |
715785.82 |
35.90 |
United Kingdom |
142386.54 |
7.14 |
Russia |
45673.77 |
2.29 |
Turkey |
28414.61 |
1.43 |
Other countries |
44554,93 |
2.23 |
Total |
1993767.56 |
100.00 |
Table 2. Conversion of lignin in different solvents [46].
Feed |
Solvent |
Catalyst |
Reaction conditions |
Major product |
Major product yield (%) |
Ref. |
|
Temp. (°C) |
Time (h) |
||||||
Poplar wood sawdust |
Alkaline aqueous solution |
NiAl alloy |
220 |
3 |
Aromatic monomers |
76 |
[ 47] |
Kraft lignin |
Alkaline water |
Ni/ZSM-5 zeolite |
200 |
4 |
Monomers |
17 |
[48] |
Birch sawdust |
water |
Pd1Ni4/MIL-100(Fe) |
130-180 |
6 |
Phenol+ acetophenone |
56.3 |
[49] |
Organosolv lignin |
Methanol |
Cu20PMO |
300 |
24 |
Liquid phase products |
26.3 |
[50] |
Hydrolyzed lignin |
Methanol |
PD/C, CrCl3 |
300 |
4 |
Monomers |
60-86 |
[51] |
Asian lignin |
Ethanol |
Pt/C |
350 |
2/3 |
Lignin-oil |
77.4 |
[52] |
Kraft lignin |
Isopropanol |
Ni-Cu/H-Beta |
330 |
3 |
cycloalkanes |
40.39 |
[53] |
beech |
Methanol |
W/C |
200 |
6 |
Lignin. Oil |
56.4 |
[54] |
Birch lignin |
Aqueous solution |
Ru / Nb 2 O 5 |
250 |
20 |
Liquid hydrocarbons |
35.5 |
[55] |
Enzymatic lignin |
Water |
Ru / Nb 2 O 5 |
250 |
20 |
Hydrocarbons |
99.6 |
[56] |
Alkali lignin |
Ethanol |
Ni 2 P/SiO 2 + CuMgAl 2 O 3 |
340 |
4 |
Monomers |
53 |
[57] |
Corn stover lignin |
N-octane |
Ru/Al 2 O 3 + Hf (OTf) |
250 |
4 |
Hydrocarbons |
>30 |
[58] |
Table 3. Lignocellulosic material composition [65].
Lignocellulosic Material |
Cellulose (%) |
Hemicellulose (%) |
Lignin (%) |
Pineapple crown (leaves) |
45.53 ± 1.17 |
21.88 ± 0.22 |
13.88 ± 1.70 |
Pineapple peel |
40.55 ± 1.02 |
28.69 ± 0.35 |
10.01 ± 0.38 |
Pineapple core |
24.53 ± 1.68 |
28.53 ± 1.37 |
5.78 ± 0.429 |
Pineapple stubble |
32.2 |
21 |
2.83 |